# Post-Traumatic Stress Disorder Trajectories the Year after COVID-19 Hospitalization

**DOI:** 10.3390/ijerph19148452

**Published:** 2022-07-11

**Authors:** Riccardo Serra, Cristian Borrazzo, Paolo Vassalini, Chiara Di Nicolantonio, Alexia E. Koukopoulos, Cecilia Tosato, Flavio Cherubini, Francesco Alessandri, Giancarlo Ceccarelli, Claudio Maria Mastroianni, Gabriella D’Ettorre, Lorenzo Tarsitani

**Affiliations:** 1Department of Human Neurosciences, Policlinico Umberto I, Sapienza University of Rome, Viale dell’Università 30, 00185 Rome, Italy; riccardo.serra@uniroma1.it (R.S.); chiadin@gmail.com (C.D.N.); alexiakoukopoulos@gmail.com (A.E.K.); 2Department of Public Health and Infectious Diseases, Policlinico Umberto I, Sapienza University of Rome, Piazzale Aldo Moro 5, 00185 Rome, Italy; cristian.borrazzo@uniroma1.it (C.B.); paolo.vassalini@uniroma1.it (P.V.); ceci.tosato@gmail.com (C.T.); cherubini.698825@studenti.uniroma1.it (F.C.); giancarlo.ceccarelli@uniroma1.it (G.C.); claudio.mastroianni@uniroma1.it (C.M.M.); gabriella.dettorre@uniroma1.it (G.D.); 3Intensive Care Unit, Department of General and Specialist Surgery “Paride Stefanini”, Policlinico Umberto I, Sapienza University of Rome, Piazzale Aldo Moro 5, 00185 Rome, Italy; francescoalessandri1@yahoo.it

**Keywords:** PTSD, COVID-19, longitudinal study, 1-year follow-up, treatment

## Abstract

**Background**: Coronavirus disease (COVID-19) hospitalization has been related to Post-Traumatic Stress Disorder (PTSD). Available information is limited by insufficient follow-up and lack of longitudinal studies. Baseline factors (e.g., sex; obesity) have been related to PTSD, but post-hospitalization factors have not been studied. **Objective**: This study aimed to analyse prevalence, baseline, post-discharge factors and possible clinical courses of PTSD after hospitalization for COVID-19. **Method**: 109 patients (94.7% of the original sample) completed a programme of three follow-up telephone assessments during the year following hospitalization. Data included clinical and sociodemographic factors as well as psychometric tools assessing PTSD, social support, and perception of threat to life (PTL). Mixture model analysis was performed to study the longitudinal course of PTSD symptoms. Chronic (>6 months) PTSD predictors were also analysed. **Results**: 1-year PTSD period prevalence was 23.9%, peaking at six months; 11% of the patients suffered chronic PTSD. Pre- and post-hospitalization factors influenced the onset and course of PTSD over time. These included working status, PTL, and lack of social support. Interestingly, obesity, pulmonary diseases and family cluster infection seem specifically related to PTSD following COVID-19. Inversely, clinical interventions, older age and male gender were protective. **Conclusions**: PTSD following COVID-19 hospitalization is common. The analysed demographic, social, clinical, and psychological factors predict PTSD symptomatology over time and can modify odds of a chronic course. Clinicians could better identify cases at risk of a chronic PTSD course. Finally, treatment as usual appeared related to a better outcome and should be proposed to patients with PTSD.

## 1. Introduction

For almost two years, the world has been facing the severe acute respiratory syndrome coronavirus 2 (SARS-CoV-2) and the dramatic consequences of the Coronavirus Disease 19 (COVID-19) [1]. In March 2020, Italy was caught off guard and suffered the second largest outbreak of SARS-CoV-2 after China [2]. Since then, there have been other outbreaks worldwide with shocking health and social consequences (e.g., India) [3]. Frontline healthcare providers and patients involved in such life-threatening situations experienced enduring, high levels of stress, with possible consequent mental health conditions [4]. Increasing evidence shows a causal link between hospitalization for severe COVID-19 and the development of mental disorders, especially Post-Traumatic Stress Disorder (PTSD) [5,6].

PTSD can result from exposure to traumatic, overwhelming experiences which are perceived as life-threatening [7]. Its manifestations include intrusive memories, flashbacks and dreams of the event, avoidance of trauma-related cues, as well as mood, dissociative, sleep and cognitive symptoms [8]. People suffering from PTSD experience disability, occupational and educational problems, poorer social support and relational difficulties [9,10,11]. Typically, PTSD has an acute onset up to 6 months after the traumatic experience, and a chronic course is frequent. One-third of patients achieve remission within 12 months, while one-third remain symptomatic for up to 10 years [12].

Regardless of their lethality, research shows that several infective diseases can affect the mental health of the infected person [13]. Indeed, studies on other epidemics (e.g., Ebola, Severe Acute Respiratory Syndrome 1) reported long-term prevalence of PTSD in the 22–48% range [14,15]. Similarly, a recent meta-analysis on infectious disease outbreaks showed that about three out of ten who survived other coronavirus infections experienced PTSD symptoms during the outbreak [16]. More specifically, up-to-date evidence on COVID-19 shows a PTSD prevalence from 9.5% to 30.2% after hospitalization [17,18,19]. However, these studies were conducted up to 4 months after discharge and evidence of late-onset and long-term course of COVID-19-related PTSD is not yet available. Such information could be of clinical interest, also considering that 25% of the subjects will experience a delayed PTSD onset more than six months after the traumatic experience [12].

Factors such as female gender, younger age, lower level of education, lower socioeconomic status, being separated, divorced or widowed, history of mental disorder and poor social support have been shown as risk factors for PTSD after exposure to trauma [20,21,22]. Interestingly, other factors were specifically linked to PTSD after hospitalization for COVID-19. Obesity, chronic pulmonary diseases, death of a family member, and dyspnea during COVID-19 appear to be specific PTSD risk factors after COVID-19 infection [17,18,23]. 

The reduction in social support consequent to the imposed social restrictions has also been linked to worse mental health outcomes and increased prevalence of depression and anxiety [24,25] and COVID-19 social restrictions might have affected patients in the post-hospitalization period. Some patients might receive specialized care to counter PTSD symptoms after discharge, but variables like *social support* and *clinical interventions during follow-up* are more dynamic in nature as compared to obesity and other chronic health conditions. Available follow-up studies, with only one assessment after a period of time, do not allow us to understand the evolution of the picture over time. Longitudinal data instead, with multiple assessments over time, could help in understanding the causality chain and classify cases based on the course of their condition. This could allow clinicians to identify cases at risk of chronic PTSD after COVID-19 hospitalization and implement targeted interventions. 

We take on these points by means of a longitudinal observational study which included previous research on 3-month prevalence and risk factors for PTSD after COVID-19 hospitalization [17]. We studied 1-year prevalence, risk factors and trajectories of PTSD. To elaborate, the aims of the study were: (1) to estimate the 1-year prevalence of PTSD in COVID-19 survivors after hospitalization; (2) to study baseline and midway factors influencing PTSD course; (3) to investigate possible clinical courses; and (4) to highlight risk factors for the worst longitudinal trajectories (no symptoms over time vs. chronic disorder).

## 2. Materials and Methods

### 2.1. Subjects

This study is the result of a collaboration between the Department of Public Health and Infectious Diseases and the Department of Human Neurosciences of the “Umberto I” General Hospital of the Sapienza University of Rome. “Umberto I” is a public, 1200-bed University Hospital, with a broad catchment area of 600,000–1,200,000 people [26]. During the hot phase of the pandemic, it was designated as one of five COVID-19 hospitals in the metropolitan area of Rome. Between March 1st and April 30th, patients with confirmed COVID-19 cases who were hospitalized in the Division of Infectious Diseases were consecutively recruited before discharge to home care. Inclusion criteria were: >18 years of age, absence of clinically significant cognitive impairment, sufficient Italian language skills, and no ICU treatment in the past 24 months (the last was applied to exclude the possibility of previous hospital-related traumatic events). Only patients discharged to home care were included. All participants gave their written informed consent to participate. 

### 2.2. Procedures

One or two days before discharge, the infectiologists working in the COVID-19 unit approached patients and screened them for inclusion in the study. Those who met the inclusion criteria were asked to participate after a thorough explanation of the study. Sociodemographic and past and present clinical data were gathered from clinical charts. A series of three follow-up telephone calls were carried out by trained clinical raters at three, six, and twelve months after hospital discharge. Raters first called the included patients and, if there was no answer, they would send a text message explaining the reason for the call and asking for a telephone appointment. This procedure was repeated up to three times in a time span of one month for each follow-up. Calls included an ex novo explanation of the study, a verbal restatement of the participant’s consent. Tools evaluating PTSD symptoms and stress vulnerability, including a subscale on social support, were then administered. The last assessment also included a retrospective evaluation of the patients’ perception of threat to life (PTL) during the hospitalization. In case of clinical signs or symptoms of a mental disorder, patients were referred to either the Hospital Psychiatric Outpatient Service or to other mental healthcare providers in the community as appropriate. The study received the approval of the Local Ethical Committee of Sapienza University of Rome (num. Rif. 109/2020). 

### 2.3. Measures

#### 2.3.1. Sociodemographic and Clinical Variables

Sociodemographic characteristics were extracted from the patient’s clinical charts. They included sex (m/f), age (< or >65 years), highest level of education (elementary-, middle-, high-school, or higher education). Relevant COVID-19-related variables were gathered using both clinical charts and telephone interviews. They included length of hospital stay (number of days), undergoing intensive treatments (yes/no), death of a hospital roommate (yes/no), family cluster (i.e., multiple family members being infected together at the same time) and COVID-19-related death of a family member (yes/no), modifications in the occupational status due to the pandemic (0 to 30% salary reduction/salary reduction >30%/fired or unemployed), and presence of any mental and physical comorbidity (both yes/no). Obesity (yes/no) and chronic pulmonary disease (yes/no) were analysed separately from other chronic medical conditions. During every telephone contact, patients were also asked if they started any medication or psychological therapy due to psychological symptoms which ensued after hospitalization.

#### 2.3.2. Post-Traumatic Stress Disorder

In order to evaluate the presence of PTSD, we used the Post-Traumatic Stress Disorder CheckList (PCL-5) from the fifth version of the Diagnostic and Statistical Manual of Mental Disorders (DSM-5), a 20-item 5-point Likert scale (0 = “Not at all” to 4 = “Extremely”), assessing DSM-5 PTSD [27]. It was created as an update to the Post-Traumatic Stress Disorder Checklist, which is a reliable tool used worldwide for the screening of PTSD. According to the National Center for PTSD, items rated as 2 (“Moderately”) and higher are considered symptoms endorsed and a PTSD diagnosis can be assigned applying the DSM-5 diagnostic rule: one B item (intrusion symptoms associated with the traumatic event e.g., intrusive memories, dreams, flashbacks), one C item (avoidance of trauma-related stimuli), two D items (cognitive and mood symptoms) and two E items (alterations in arousal and reactivity) (8). A PCL-5 total >30 can also be used in order to indicate the presence of a clinically relevant PTSD symptomatology [28]. 

#### 2.3.3. Perceived Lack of Social Support

The Stress-related Vulnerability Scale (SVS) [29] measures subjective stress vulnerability over time. It is a self-reported rating scale composed of nine items, scored on a 4-point Likert scale (0 = “Not at all” to 3 “A lot”). It yields scores on three subscales of three items: named “Tension”, “Demoralization”, and “Lack of Social Support”. Higher scores indicate greater subjective stress-related vulnerability. The SVS provided evidence of convergent validity, satisfactory internal consistency, high test–retest reliability and sensitivity to change. It was fruitfully used in clinical settings [30,31]. For the scope of the present research, and to avoid collinearity with the PCL-5, only the “Reduced Social Support” has been included in the multivariate analyses. Full scores are presented with descriptive statistics.

#### 2.3.4. Patient’s Perception of Threat to Life during the Hospitalization

In order to quantify PTL during hospitalization, we retrospectively administered a validated, brief, self-reported measurement assessing these dimensions in an acute care setting [32]. It was successfully used in order to predict the development of PTSD after hospitalization for acute coronary syndrome [33]. This is a 7-item 4-point Likert scale (0 = “Not at all” to 4 = “Extremely”) assessing different aspects of fear felt in relation to the experience (e.g., “I was afraid I would die”, “I felt vulnerable”). The resulting total score was used as a continuous variable. 

### 2.4. Statistical Analysis

Descriptive statistics of quantitative variables were reported either as continuous variables including a median with interquartile range (IQR, 25 to 75%) or a mean with standard deviation (±SD). Simple frequencies and percentage (%) were used for categorical variables. A Kolmogorov–Smirnov test was used to verify values’ normal distribution. Normal variables were analysed using an independent sample Welch’s *t*-test, while the Wilcoxon signed-rank test was used for variables not normally distributed. A Chi-square test or Fisher’s exact test was used for categorical variables, as appropriate. 

A Welch’s Analysis of Variance (ANOVA) was used to assess group differences for continuous outcomes. A Welch’s *t*-test assuming unequal variances was used for post hoc comparisons. The same test was also used to compare patients divided into two groups: PTSD and no PTSD.

A two-sided *p*-value test < 0.05 was considered statistically significant. Prior to analysis, data were screened for missingness and participants who did not complete every assessment were excluded. As a few missing data were deemed to be inevitable with this study design, no attempt was made to impute missing values; all collected data were included in the analyses. Time was coded as T3, T6, T12 at the 3-, 6-, and 12-month follow-ups, respectively. It was imputed as a continuous variable in the statistical analysis, assuming that the change in the response variable is linear over time. When imputed as a categorical variable, “time” did not significantly influence the model. The “before–after plot” shows the clinical trajectories of patients who presented with PTSD at any time during follow-up.

Longitudinal data were analysed by jointly considering all three follow-up measurements (i.e., 3-, 6- and 12-month) and using a random intercept model to accommodate within-subject correlation over time. Matched bivariate analysis was conducted in a stepwise, conditional, logistic regression model.

In the multivariate analysis, the formula for calculating a *z*-score is *z* = (*x* − *μ*)/*σ*, where *x* is the raw score, *μ* is the population mean and *σ* is the population standard deviation. Odds ratios (ORs) and 95% confidence intervals (95% CIs) were calculated for all associations. All data were analysed using commercially available statistical software packages (SPSS Statistics for Mac, 22.0; IBM Corp., Armonk, NY, USA). Methods of unsupervised clustering, statistical tests and regression analyses were implemented utilizing R statistics software.

## 3. Results

A total of 183 patients were discharged from the hospital unit to home care in the selected recruitment period (i.e., between March and April 2020) and were considered for participation. Of these, 25 did not fulfill the inclusion criteria and 2 refused to participate in the study. Forty-one patients did not complete the first follow-up assessment and were excluded from the sample. The reasons for not completing the first assessment were: telephone number was unavailable or not working (*n* = 20); telephone number worked but patients did not answer (*n* = 20); patient was deceased (*n* = 1). Of the 115 patients who completed the first telephone interview, a total of *n* = 109 (94.7%) completed all the assessments and were included in the present analysis. This sample proved strongly representative of the original group with 98% preserved statistical power. Patients lost during the recruitment phase and those included presented with no difference in age (*p* = 0.20) or sex (*p* = 0.3). The PTL total score was normally distributed in our sample.

### 3.1. Point and Period Prevalence of PTSD at the 12-Month Follow-Up

At the 12-month follow-up, 12.7% (*n* = 14) of the sample received a PCL-5-based diagnosis of PTSD. 

A total of *n* = 26 patients received a PCL-5-based diagnosis of PTSD at some point during follow-up for a total period prevalence of 23.9%. Furthermore, prevalence peaked at 6 months and decreased after.

### 3.2. Sample Description and Group Comparison

Table 1 shows the demographic, social, clinical, and psychological characteristics of the sample, as well as a sub-group comparison between patients who received a diagnosis of PTSD at a follow-up and those who did not.

The majority of the sample was male. The mean age was 57, with roughly two-thirds of the patients (61%) being over 65. A total of 55% were hospitalized in relation to a household infection and one patient in ten (9%) experienced the loss of a family member due to that COVID-19 infection. The average length of hospitalization was 15 days, and during the hospitalization, 16% of the patients witnessed a roommate’s death. Once discharged, about one in five were either fired from work (6%) or underwent a significant reduction of income (12%). The mean score for perceived lack of social support was four out of five in 2 follow-up assessments and 15% of the patients sought help for their mental condition after discharge. Obesity (*p* < 0.01), PTL (*p* = 0.003), lack of social support (*p* < 0.001 in every assessment) and clinical interventions at T6 (*p* = 0.019) were significantly different between patients with and without PTSD at any assessment. More specifically, out of the 6 patients who received a PTSD diagnosis at T6 and sought specialized clinical help, 5 achieved recovery at the following assessment. In order to test the hypothesis that PTL was higher in patients with risk factors for COVID-19 death, we compared a group with obesity and chronic pulmonary diseases with the rest of the patients. The results show that the first group had significantly higher PTL (mean score = 22; range = 18–28) than the group with no risk factors (mean score = 15; range = 3–25; *p* < 0.001).

### 3.3. Mixture Model Predicting the Longitudinal Course of PTSD

Table 2 shows results from the mixed model with a random intercept of factors analysis. The results indicate that the categorical variables male gender and age **≥** 65 were related to a lower PCL-5 score over time. Oppositely, the dichotomous variables pre-existing mental disorders, obesity, chronic pulmonary disease, family cluster, lack of social support and the continuous variables PTL predicted higher PCL-5 over time.

Possible PCL-5 total score longitudinal trajectories are presented in Figure 1, first diagram. The second diagram (Figure 2) shows the clinical trajectories of patients who presented with PTSD at any time during follow-up. Specifically, 12 patients had an early PTSD onset, with the vast majority (*n* = 8; 66%) showing an enduring PTSD course. The other 10 patients had a PTSD onset between 3 and 6 months after hospital discharge, but only 4 of them had PTSD at the following follow-up. Finally, a total of 4 patients underwent a delayed PTSD onset after 6 months.

### 3.4. Multivariate Logistic Regression Model Predicting Chronic PTSD

Chronic PTSD was defined as having an early onset (i.e., at T3 or T6) and being still positive at T12. Predictors of chronic PTSD highlighted in univariate analysis were included in our multivariate model (results from the univariate analysis are available upon request). The results from the multivariate analysis are shown in Table 3. Patients with an anamnesis of previous psychiatric diagnosis (*p* = 0.027), obesity (*p* = 0.022), or a chronic pulmonary disease (*p* < 0.001) had 3.3-, 3.9- and 7.7-folds increased risk of undergoing chronic PTSD, respectively. Family cluster (OR = 3.2; *p* = 0.011) and PTL (OR = 1.1; *p* < 0.001) were also significant over and above other variables in the model. Of the midway variables, assessed at follow-up, working status (OR= 4.370; *p* = 0.018) and lack of social support (OR=1.6; *p* = 0.033) showed a direct relation with chronic PTSD. On the other hand, patients who benefitted from specialized interventions during follow-up had a reduced odds ratio (OR) of chronic PTSD of 0.66 (*p* = 0.009).

## 4. Discussion

For one year, beginning with the Italian COVID-19 outbreak, we studied PTSD in a sample of infected patients hospitalized in a non-intensive care unit of a large university hospital in Rome. Through multiple follow-up assessments, we gathered data on the time of onset, period prevalence and the clinical course of PTSD in these patients. Our results indicate that, during the year following discharge, one patient out of four undergoes an onset of PTSD. In line with the epidemiology after a traumatic event [12], prevalence peaks at six months and then declines over time. Still, some patients undergo a PTSD onset after more than six months. The results indicate that half of the patients diagnosed with PTSD will undergo a transient disorder, with remission at the following assessment. The other half will instead face the chronicization of their condition. Demographic, social, clinical and psychological factors impacted in the onset and duration of PTSD in this sample.

### 4.1. Demographic Factors

In line with existing literature, being female proved predictive of PTSD during follow-up [34] and was correlated to the longitudinal trend of PLC-5 total score. Hence, women were more likely to experience PTSD symptoms over time. Yet, our study shows that gender was not multivariately associated with having chronic PTSD. A low level of education has been reported to predict PTSD in other settings [35], but this finding was not confirmed in the present case. Inversely, being older than 65 had a protective role in the development of a diagnosis of PTSD and its symptoms over time, confirming previous findings on the general pathogenesis of PTSD [36]. Age, however, was not multivariately associated with chronic PTSD.

### 4.2. Social Factors

The socioeconomic consequences of the pandemic also emerge from our data. Many patients were unemployed, fired or impoverished during the year following their hospitalization. Conversely to those having a job (even if with a reduced income), patients who were unemployed or fired had a higher risk of receiving a PTSD diagnosis during follow-up. This result confirms previous literature on PTSD [37]. A novel finding of our research is that being unemployed/having been fired before a follow-up is multivariately associated with chronic PTSD, with four to five times higher odds than being employed. Perception of social support was low across most assessments throughout the entire sample, underlining the difficult moment we faced as a society. Social support is crucial for mental health [38], and its bond with PTSD was further stressed in a recent study showing a bidirectional relation between PTSD and social isolation over time [39]. The necessary social restrictions imposed during the beginning of the pandemic definitely influenced the perception of social support within the entire sample. Specifically, people reporting a severe lack of social support were at high risk of developing PTSD and undergoing a chronic clinical course. 

### 4.3. Clinical Factors

More than half of our patients underwent a hospitalization of 15 days or more. Long hospitalizations are likely related to two factors: the absence of effective treatments at the time of the outbreak and the difficulties of discharging clinically remitted patients who tested positive in nasal swab tests. Indeed, this was a common case during the first wave of the pandemic, with some reported cases hospitalized for more than 50 days [40]. However, the length of hospitalization showed no relation to PTSD in our sample. Counter-intuitively, the use of more intense and potentially traumatizing treatments (e.g., mechanical ventilation) was not related to PTSD in our sample. In line with existing literature on COVID-19, people with a history of psychiatric diagnosis, obesity and chronic pulmonary diseases were at higher risk of developing the disorder during follow-up and had three to seven times the odds of undergoing a chronic PTSD course than their counterparts. Regarding these specific factors, in a previous report of earlier follow-up data [17], we hypothesized that people with risk factors linked to COVID-19 death had a higher perception of threat to life (PTL) and therefore were more at risk of PTSD. In the present article, this hypothesis proved valid. Indeed, people with known factors related to death due to COVID-19 (i.e., obesity and chronic pulmonary disease) had higher PTL than healthy individuals. Further research could consider these as factors related to higher PTSD due to similar mechanisms and test the possible mediating role of PTL in the relation between obesity and chronic pulmonary diseases and PTSD following COVID-19. It is encouraging that psychiatric interventions were associated with a significant reduction in PTSD prevalence. Moreover, those who underwent specialized treatment after hospitalization had reduced odds of undergoing chronic PTSD over and above all the variables included. Lack of time between discharge from the hospital and the 3-month follow-up likely determined the non-significant impact of psychiatric treatment at the first follow-up assessment. Indeed, finding a specialist and starting therapy during the strict lockdown phase was difficult, and the time for drug titration and/or any psychological interventions was likely insufficient.

### 4.4. Psychological Factors

Most of our patients were infected in the context of a household infection, and grimly, many lost one member of their family due to COVID-19. Household infections are reported as the most common way of COVID-19 transmission [41], and in our sample this condition resulted to be predictive of a PTSD diagnosis during follow-up. It is hard to disentangle the many factors possibly involved in this finding. Surely, a role was played by the additional PTL patients who experienced knowing that the life of a loved one was at risk. Indeed, our models confirm literature indicating PTL as a key component in the etiology and maintenance of PTSD [42]. Importantly, PTL was also multivariately predictive of chronic PTSD.

### 4.5. Limitations

The results of our study have to be interpreted in the light of at least three main limitations, potentially reducing the generalizability of the finding. First of all, the sample was limited in size and it was entirely from a single hospital in a western capital city. Still, the rate of attrition at follow-up was extremely limited, guaranteeing internal representativity. Secondly, our study involved a follow-up telephone assessment and a checklist-based PTSD diagnosis. Such methods are surely less reliable than in-person structured interviews but had to be chosen in consideration of the unprecedented and dangerous situation. Thirdly, PTL was retrospectively assessed after hospital discharge. Surely, assessing PTL during the hospitalization would have reduced memory biases. In addition to these, the limited number of patients with morbid obesity and chronic pulmonary diseases might represent a limitation of this study.

## 5. Conclusions

PTSD following hospitalization for COVID-19 is common. Prevalence peaks in the first 6 months, with few patients undergoing a late onset. Demographic, social, clinical, and psychological factors have been highlighted as risk factors for PTSD onset, trajectories and clinical courses. The present evidence might help clinicians in identifying cases at risk of chronic PTSD in order to better direct screening and treatment efforts. Finally, treatment as usual was associated with a decreased prevalence of PTSD and should be proposed to patients with PTSD after COVID-19 hospitalization.

## Figures and Tables

**Figure 1 ijerph-19-08452-f001:**
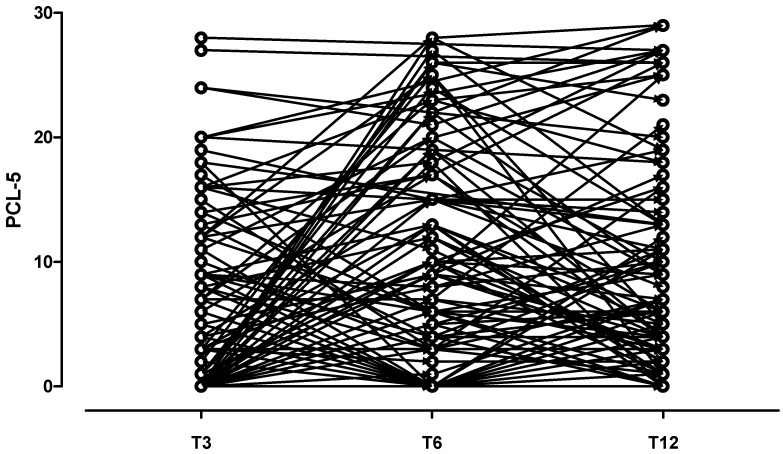
PCL-5 total score trajectories in patients without PTSD.

**Figure 2 ijerph-19-08452-f002:**
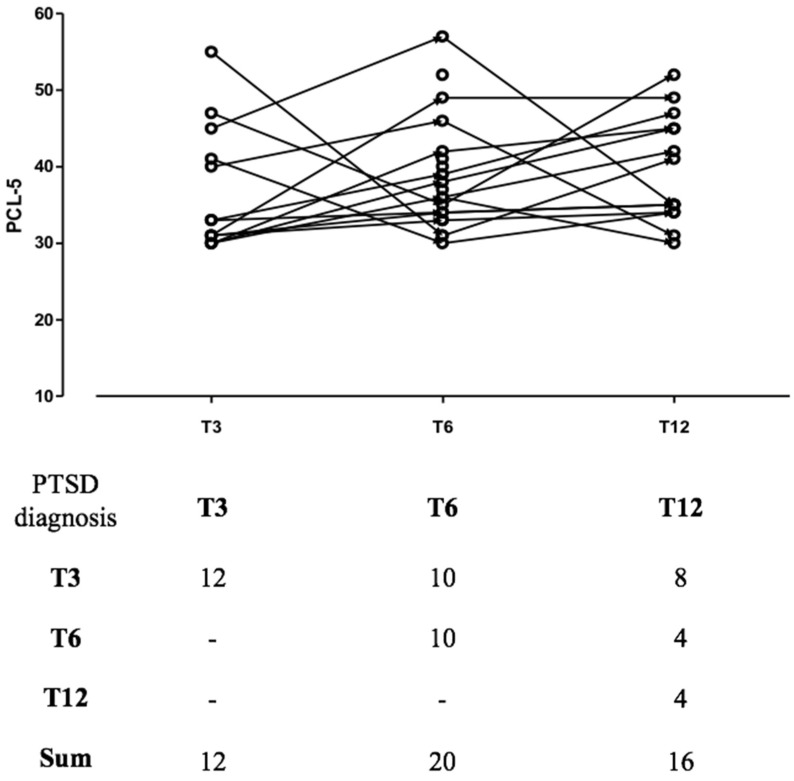
PCL-5 total score trajectories in patients with PTSD and the total amount of patients with PTSD at different follow-up assessments.

**Table 1 ijerph-19-08452-t001:** Demographic, clinical and treatment characteristics in the whole sample and in the 12-month period PTSD subgroups.

Sociodemographic	Sample (*n* = 109)	PTSD (*n* = 26)	No PTSD (*n* = 83)	*p*-Value
**Male, *n* (%)**	63 (61)	6 (23)	57 (69)	
**Female, *n* (%)**	46 (42)	20 (77)	26 (31)	**<0.001**
Age, median (IQR)	57 (45–66)	57 (51–62)	58 (47–66)	0.320
**≥65 years, *n* (%)**	30 (28)	3 (12)	27 (32)	**0.042**
Education				
Elementary school, *n* (%)	9 (8)	2 (8)	7 (8)	1.000
Middle school, *n* (%)	13 (12)	4 (15)	9 (11)	0.585
High school, *n* (%)	43 (42)	8 (31)	35 (42)	0.319
Higher education, *n* (%)	44 (40)	12 (46)	32 (38)	0.469
Days of hospital stay; Median (IQR)	15 (9–24)	14 (9–20)	16 (10–24)	0.222
Hospital stay (≥15 days) *n* (%)	57 (52)	12 (46)	45 (54)	0.478
Intensive treatment, *n* (%)	28 (26)	4 (15)	24 (29)	0.156
Past psychiatric diagnosis, *n* (%)	9 (8)	4 (15)	5 (6)	0.145
**Obesity (BMI > 30), *n* (%)**	5 (5)	3 (12)	2 (2)	**0.030**
Chronic pulmonary diseases, *n* (%)	15 (14)	6 (23)	9 (11)	0.124
Death of a roommate, *n* (%)	17 (16)	3 (12)	14 (17)	0.544
Death of a family member, *n* (%)	10 (9)	2 (8)	8 (10)	0.763
**Family cluster (i.e., being infected together with other family members)** **, *n* (%)**	60 (55)	19 (73)	41 (49)	**0.033**
**Chronic medical illness, *n* (%)**	60 (55)	8 (31)	52 (63)	**0.004**
Working status				
Stably employed, *n* (%)	82 (75)	16 (62)	66 (79)	0.082
**unemployed** **, *n* (%)**	8 (7)	6 (23)	2 (2)	**<0.001**
≥30% reduction, *n* (%)	13 (12)	2 (8)	11 (13)	0.492
fired, *n* (%)	6 (6)	2 (8)	4 (5)	0.567
**PTL, mean ± SD**	16.8 ± 6.2 (3–28)	21.5 ± 5.8 (18–28)	15 ± 5.9 (3–25)	**<0.001**
**SVS lack of social support, median (IQR)**				
**T3**	4 (0–5)	5 (3–6)	1 (0–2)	**<0.001**
**T6**	3 (0–4)	4 (3–6)	1 (0–2)	**<0.001**
**T12**	4 (0–5)	5 (3–6)	1 (0–2)	**<0.001**
**Clinical interventions during follow-up, *n* (%)**				
T3	16 (15)	3 (12)	13 (16)	0.620
**T6**	16 (15)	7 (27)	9 (11)	**0.046**
T12	16 (15)	1 (4)	15 (18)	0.079

Note: PTSD = post-traumatic stress disorder; BMI = body mass index; SVS = Stress-related vulnerability scale; PTL = perception of threat to life; IQR = interquartile range. Significant variables are highlighted in **bold** and gray background.

**Table 2 ijerph-19-08452-t002:** Random intercept model prediction of PCL-5 total score over time.

Variables	β	±SE	*t*	*p*-Value	95% CIs
**Gender, male**	**−5.6**	**1.3**	**−3.89**	**<0.001**	**(−7.5, −2.6)**
**Age ≥ 65**	**−4.9**	**1.6**	**−3.03**	**0.003**	**(−7.7, −1.7)**
Length of hospital stay (≥15 days)	1.2	1.3	1.11	0.266	(−1.1, 4.6)
Intensive care treatment	0.6	1.7	0.28	0.783	(−2.8, 3.4)
**Previous psychiatric diagnosis**	**4.2**	**2.4**	**3.1**	**0.021**	**(2.2, 10)**
**Obesity (BMI > 30)**	**1.3**	**3.4**	**2.3**	**0.043**	**(0.1, 5.6)**
**Chronic pulmonary diseases**	**7.8**	**1.7**	**4.68**	**<0.001**	**(4.5, 11.3)**
Death of a roommate	1.2	1.6	0.73	0.468	(−2, 4.5)
Death of a family member	−2.1	2.1	−1.0	0.316	(−6.1, 2.2)
**Family cluster (i.e., being infected together with other family members)**	**3.0**	**1.2**	**2.5**	**0.015**	**(0.6, 5.4)**
Other chronic medical Illness	1.2	1.3	0.94	0.340	(−1.3, 3.8)
Education					
Elementary school	1.7	1.2	0.9	0.455	(−0.9, 12.1)
Middle school	−0.8	1.1	−2.2	0.585	(−4.2, 3.3)
High school	1.1	2.0	1.8	0.319	(−1.8, 6.2)
Higher education	1.0	2.2	3.1	0.469	(−0.8, 6.7)
Working status					
**Employed**	**−1.4**	**1.4**	**−3.2**	**0.044**	**(−3.8, 4.4)**
**Unemployed/fired**	**3.2**	**1.1**	**1.2**	**0.012**	**(2.4, 6.6)**
**SVS lack of social support**	**4.1**	**2.6**	**−4.1**	**<0.001**	**(3.4, 10.2)**
Clinical interventions during follow-up	−1.2	2.2	−3.2	0.169	(−4.2, 3.2)
**PTL**	**7.2**	**2.4**	**−3.8**	**<0.001**	**(4.2, 12.4)**

Note: PCL-5 = post-traumatic stress disorder checklist for DSM-5; BMI = body mass index; PTL = perception of threat to life; β = regression coefficient of the linear mixed model with random intercept; CIs = confidence intervals. Significant variables are highlighted in **bold** and gray background.

**Table 3 ijerph-19-08452-t003:** Multivariate logistic model predicting chronic PTSD.

	Estimate	Standard Error	Odds Ratio	z	Wald Statistic	*p*-Value	VS-MPR *	CI Lower Bound	CI Upper Bound
Gender	−0.72	0.426	0.486	−1.692	2.782	0.091	1.69	0.211	1.121
Age (≥65 years)	−0.8	0.569	0.451	−1.400	1.493	0.161	1.25	0.148	1.375
Hospital stay (≥15 days)	0.11	0.425	1.118	0.260	0.060	0.793	1.0	0.486	2.570
**Previous psychiatric diagnosis**	**1.21**	**0.549**	**3.363**	**2.209**	**4.814**	**0.027**	**3.75**	**1.146**	**9.864**
**Obesity (BMI > 30)**	**1.36**	**0.593**	**3.899**	**2.293**	**3.405**	**0.022**	**4.41**	**1.219**	**12.473**
**Chronic pulmonary diseases**	**2.043**	**0.492**	**7.714**	**4.150**	**16.522**	**<0.001**	**4.99**	**2.940**	**20.244**
Education									
Elementary school	2.66	0.92	10.1	1.08	1.288	0.092	1.40	0.95	16.8
Middle school	−0.96	0.7	0.5	−0.38	0.116	0.706	1.30	0.71	3.2
High school	1.1	0.6	2.6	0.61	0.254	0.545	1.60	0.5	4.1
Higher education	1.2	0.7	2.6	0.61	0.244	0.545	1.70	0.8	4.2
Working status									
Employed	−0.64	0.614	0.527	−1.044	0.917	0.296	1.02	0.158	1.755
**Unemployed/fired**	**1.475**	**0.623**	**4.370**	**2.368**	**4.603**	**0.018**	**5.1**	**1.289**	**14.814**
Death of a roommate	0.1	0.505	1.105	0.197	0.031	0.843	1.0	0.411	2.972
Death of a family member	0.586	0.601	1.796	0.975	0.576	0.330	1.01	0.553	5.832
Invasive treatment	−0.116	0.598	0.891	−0.193	0.034	0.847	1.0	0.276	2.877
**Family cluster**	**1.7**	**0.42**	**3.220**	**1.430**	**3.540**	**0.011**	**2.4**	**1.322**	**6.42**
**PTL**	**0.14**	**0.40**	**1.147**	**3.456**	**11.563**	**<0.001**	**9.5**	**1.061**	**1.24**
**Clinical interventions during follow-up**	**−1.2**	**0.32**	**0.660**	**2.594**	**4.350**	**0.009**	**8.3**	**0.2**	**0.88**
**SVS lack of social support**	**2.71**	**0.83**	**1.62**	**0.57**	**0.242**	**0.033**	**1.0**	**1.1**	**2.1**

Note: BMI = body mass index; PTL = perception of threat to life; CI = confidence Interval. Significant variables are highlighted in **bold** and gray background. * Vovk-Sellke Maximum p -Ratio: Based on the *p*-value, the maximum possible odds in favour of H₁ over H₀ equal 1/(−e p log(p)) for *p* ≤ 0.37.

## Data Availability

The data that support the findings of this study are openly available in Mendeley Data at http://doi.org/10.17632/4fn2mhm5pr.1, accessed on 12 May 2022.

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
