# Peer review of "Post-Traumatic Stress Disorder Trajectories the Year after COVID-19 Hospitalization"

_ijerph, 2022, doi:10.3390/ijerph19148452_

Round 1

Reviewer 1 Report

Brief summary

The authors are dealing with an important and relevant topic, the relationship between COVID-19 hospitalization and the development of PTSD in a one-year follow-up study. Demographic, social, clinical and psychological data were collected from 109 patients at three time points. The authors identified the prevalence of 1-year PTSD (cca. 24%), its peak (at around 6 months) and the ratio of chronic PTSD (11%). Both risk factors and protective factors were mentioned. 

The main contribution is that the paper focuses on mental health consequences of COVID-19, and identifies the risk factors of PTSD occurance after COVID-19 hospitalization.  

The main strengths of the paper are the following:

(1) longitudinal and post-hospitalization approach,

(2) integrative point of view that takes into account several factors (e.g., demographic, social, clinical, psychological),

(3) provide a long-term course of COVID-19 related PTSD.  

Major comments and questions

The paper is well-structured, relevant for the field and cites several recent publications. However, there are some weaknesses to improve. Please, find the comments below. 

  1. 1. One of the main weaknesses is the discrepancy between the four aims (page 2, lines 89-94), and the one (and only) hypothesis (page 2 lines 70-72). Please, clarify  the specific hypotheses related to the four aims.  

  1. 2. The research aims, the hypothesis and the practical implications are somewhat divergent in the paper. What does it mean? It is unclear wether the focus is on (1) risk factors + higher PTL + more risk of PTSD (as it is stated in the hypothesis), or (2) PTSD (as implicated by the four research questions), or (3) the psychosocial and clinical aspects (as the paper suggests it as a whole). 

  1. 3. Two types of chronic medical conditions (obesity and pulmonary disease) are separated as significant factors, but the very low ratio of obesity  does not support it (5 from 109 – 5% for the total sample, 3 from 26 – 12% in the PTSD group, 2 from 83 – 2% in the No PTSD group) (see page 6, Table 1).  

  1. 4. The authors have checked the normal distribution of the values, but it is not mentioned which variables were normally distruóibuted and not. Please, add this information. 

  1. 5. It is not clear which variables were treated as categorical and continuous. Please, clarify it. (In general and especially in 3.3 section , p 5 l 232) 

  1. 6. Please, define what family cluster means (p 6, Table 1). 

  1. What is the reason to use 65 year-of-age at cut points? 

  1. 7. Conclusion made under 5.1 Demographic factors (p 11, lines 284-286) are not supported by the results. Please, clarify it. 

  1. 8. Please, reconsider Image 1 and 2 (Figures), because it does not depict the content written in the text. 

Minor comments 

  1. 9. Some results mentioned in 3.2 section (p 5, lines 229-231) do not appear in Table 1. 

  1. 10. Some sentences are hard to understand. Maybe they should be re-written (e.g., p 2, line 53 - “Evidence shows...”, p 4, l 183 - “The effect of time...”, p 5 208-210 - “A total of …") 

  1. 11. Consistent useage of abbreviations should be fine (COVID-19 or Covid-19 or covid-19, T3 or t3) 

  1. 12. What a means in Table 1 (see p values)? 

  1. 13. The heading of 5. Conclusions should be changed to 4. Discussion. (I guess one paragraph should be deleted, p 10, lines 263-266.) 

In sum, the paper can provide important contribution to a better understanding of COVID-19-related PTSD, moreover, the psycho-social and clinical factors that could be risk factors (or protective ones). However, I do not recommend it for acceptance in its current form due to some conceptual reasons (e.g. focus should be clarified, research questions and hypotheses should converge). Moreover, additional information should be provided regarding data- analysis (e.g. to list normally distributed variables, categorical and continuous variables), and the interpretation of the results should be modified at some points.  

Reviewer 2 Report

This is an interesting article that assessed a very critical issue. It is well written in general. However the discussion part is missing or at least something is not going well. Please check. Additionally, I have made several comments that need to be addressed.

Abstract

-          Too long the background. Please shorten it.

-          Give more info about the methods;

-          In the conclusions you say that … increase the odds of PTSD… but no such results are presented;

Introduction

-          Row 56. 3 out of ten;

-          Please check this article: https://www.thelancet.com/journals/lancet/article/PIIS0140-6736(21)02143-7/fulltext

-          Last paragraph of introduction: Some parts belong to methods. Please remove;

Methods

-          How did you approach the patients? Did you had a list of those who will be discharged? Did you train somebody for this?

-          Rows 138-139: Please give full name of DSM-5 (PCL-5);

Results

-          Please give more information about “46 were lost at Follow-up”. It’s not clear if no response the first time did you recall them?

-          The “More than half” is used several times. Please rephrase;

The discussion part is missing. Please check;

Conclusion part is written twice

Round 2

Reviewer 1 Report

The authors have made significant modifications to improve the quality of the paper, and they responded to my previous comments. 

Thank you.

Reviewer 2 Report

Thank you for addressing the comments appropriately. I have no more comments.